# *Leishmania* and the Model of Predominant Clonal Evolution

**DOI:** 10.3390/microorganisms9112409

**Published:** 2021-11-22

**Authors:** Michel Tibayrenc, Francisco J. Ayala

**Affiliations:** 1Maladies Infectieuses et Vecteurs Ecologie, Génétique, Evolution et Contrôle, MIVEGEC (IRD 224-CNRS 5290-UM1-UM2), Institut de Recherche Pour le Développement, CEDEX 5, 34394 Montpellier, France; 2Catedra Francisco Jose Ayala of Science, Technology, and Religion, University of Comillas, 28015 Madrid, Spain; Fjayala2018@gmail.com

**Keywords:** evolution, clonality, genetic recombination, population structure, aneuploidy, molecular epidemiology, parasite, leishmaniosis

## Abstract

As it is the case for other pathogenic microorganisms, the respective impact of clonality and genetic exchange on *Leishmania* natural populations has been the object of lively debates since the early 1980s. The predominant clonal evolution (PCE) model states that genetic exchange in these parasites’ natural populations may have a high relevance on an evolutionary scale, but is not sufficient to erase a persistent phylogenetic signal and the existence of bifurcating trees. Recent data based on high-resolution markers and genomic polymorphisms fully confirm the PCE model down to a microevolutionary level.

## 1. Introduction

The “clonality/sexuality debate” has been running among microbiologists since the early 1980s [1,2,3], thanks to the advent of reliable genetic markers, such as Multilocus Enzyme Electrophoresis* (MLEE). The issue is relevant for basic science (knowledge on the basic biology of the concerned pathogens), as well as for applied research (strain typing, epidemiological follow-up). As a matter of fact, if the concerned species is sexual (=recombining*), its multilocus genotypes* (MLGs) are unstable and vanish in the common gene pool after each recombination episode. On the other hand, if the species is clonal, recombination does not operate, and the MLGs of the species under study are stable in space and time.

In the framework of this debate, we have proposed “the clonal theory of parasitic protozoa” [4], which has been afterwards extended to other eukaryotic pathogens, then to bacteria and viruses [5].

We present here the most recent developments of this theory dealing with the *Leishmania* genus.

## 2. A Brief Recall on the Clonal Theory in Its Present Form: The Predominant Clonal Evolution Model (PCE)

Since its early developments, we have always given the same meaning to clonality: restrictions to genetic recombination impacting the population structure of the species under study. The definition, therefore, deals with the actual effects of clonality on population structure, and neither on cytological aspects of sex, nor on specific mating patterns. This definition of clonality is accepted by many, if not most, authors working on microbial biology (see for extensive references [5,6,7,8]), and on clonal metazoa [9]. Some authors do not accept this definition of clonality, and recommend distinguishing selfing/inbreeding from “true” clonality (=mitotic propagation) [9]. Other authors equate clonality with genetic monomorphism [10].

### 2.1. What Are the Units of Analysis Considered by the PCE Approach?

The first unit of analysis is the species as it is described and accepted by the experts of the field. Example: does *Leishmania infantum* undergo PCE? In a second step, intraspecific patterns are explored (microevolution*).

### 2.2. Where to Put the Cursor?

The clonality/sexuality debate has turned circles around the “amount of sexuality” (=recombination) that would “challenge” the PCE model. The problem has been worsened: (i) by incorrect definitions of the PCE model such as “strict clonality” (=no sexuality at all) [9,11,12]; (ii) by wrong assertions concerning it (“sexuality of little epidemiological and evolutionary relevance”) [13], (“sex inconsequential”) [14]. Now, since the very first statements about the clonal theory, we have constantly insisted on the fact that PCE: (i) does not say that recombination is absent, but rather, that its impact is not sufficient to break the prevalent PCE pattern; (ii) that sex may have a high epidemiological and biological relevance, but only on an evolutionary scale.

Recurrent use of imprecise, subjective terms (“frequent” genetic exchange [14]); sex “much more frequent than previously thought” [11]) has made the debate even more confusing, hence, the need for sharply defined criteria.

### 2.3. The “Clonality Threshold”: Phylogenetic Signal/Clonality Backbone/Bifurcating Trees

According to many authors, the presence of bifurcating trees and tree-like structures is incompatible with a predominant recombination pattern and is therefore a specific signature of a clonal frame or “clonal backbone” [15,16,17,18,19]. Evidencing a detectable phylogenetic signal, stable in time and in the whole ecogeographical range of the concerned species is, therefore, the main manifestation of PCE. The means to detect this signal are the classic ones used in phylogenetic analysis (bootstrap analysis*, with a threshold of 0.70, as proposed by Hillis and Bull, 1993). This criterion is all the more significant when the phylogenetic signal is reinforced by adding more data (congruence criterion): more molecular markers, more populations, more sampling places, and/or when the phylogenetic signal is corroborated by different phylogenetic and nonphylogenetic methods such as STRUCTURE [20]. This is what we have called the “clonality threshold” [6], beyond which clonality irremediably surpasses recombination, and different clonal lines will diverge from each other even more. The correlation between different kinds of genetic markers (for example, MLEE and DNA markers) corresponds to our “g test” of clonality [4]. The presence of deep phylogenies within a given species makes it possible to distinguish PCE from the so-called semiclonal/epidemic clonal model [16], a pattern of population structure in which occasional bouts of clonality appear in an otherwise recombining species. As a matter of fact, according to this model, the clonal lines are ephemeral and disappear in the recombining gene pool after at most a few decades.

### 2.4. The Near-Clade (NC) Concept

Even if bifurcating trees are clearly established by a valid phylogenetic analysis, complete clonality probably does not exist in the world of pathogens. The term “clade” is therefore inappropriate, since clades are strictly separated evolutionary units. This is why we have coined the term “near-clade” (NC) [5], which designates a stable evolutionary unit whose discreteness is somewhat blurred by occasional recombination. However, these bouts of sex are unable to erase the phylogenetic signal. 

### 2.5. Other PCE Features

Other marks of clonality comprise widespread clonal multilocus genotypes and linkage disequilibrium.

The first feature is a logical consequence of clonal propagation. If a species undergoes regular recombination, its MLGs are unstable and soon disappear in the common gene pool. Repeated MLGs should be exceptional, and statistically compatible with panmictic* expectations. If this is not the case, it is evident that clonality obtains.

Linkage disequilibrium (LD) designates the non-random association of genotypes occurring at different loci. In a sexual species, the opposite is obtained (linkage equilibrium). A statistically significant linkage disequilibrium provides classic circumstantial evidence of obstacles to recombination and, hence, to clonality. It has been used towards this goal by many authors working on pathogen population genetics (see Table 2 in [6]). It is considered as the best approach to evidence obstacles to recombination [15,21]. It has been criticized for its supposed lack of power [22]. This criticism is unjustified. If sampling is valid and avoids the Wahlund* effect, and if a sufficient range of genetic markers is used, LD is a very powerful test.

### 2.6. Side Concepts: Clonet, Russian Doll Pattern (RDP)

A “clonal genotype” (monomorphic MLG) should be actually viewed as a family of related clones. If markers with a higher resolution are used, the “clone” will prove to be genetically heterogeneous. We have forged the term of clonet to designate a clonal MLG that appears to be monomorphic with a given set of genetic markers [23] (see Figure 1).

The Russian doll model [24] states that within each of the NCs that subdivide the species under study, PCE is observed in the same way as at the level of the whole species, with widespread clonal MLGs, LD and lesser NCs (Figure 1). Such Russian doll patterns (RDPs) make it possible to distinguish PCE from two alternative models, namely the pseudo-speciation model and the progressive clonality model. In the first one, lack of recombination is observed between the NCs, while within them, genetic exchange is much more frequent [14]. The progressive clonality model infers that the frequency of genetic exchange is inversely proportional to the genetic distance that separates two given MLGs [10]. In the two cases, recombination should be abundant within each NC, and so, RDPs should not be observed.

### 2.7. Resolution of Markers and the PCE Model

This is a recurrent criticism to the PCE model. It would be artefactual and wrong due to the lack of resolution of the classic markers* (MLEE, Random Amplified Polymorphic DNA* {RAPD}, Restriction Fragment Length Polymorphism* {RFLP}). [22,25,26,27]. This criticism is invalid for two reasons: (1) lack of resolution, that is to say: lack of sufficient data, should mimic sexuality (null hypothesis) rather than clonality (working hypothesis). (2) All the recent developments of the PCE approach, starting from 2012 [5] rely on the analysis of high-resolution markers, such as microsatellites* and of genomic data (whole genome sequencing* (WGS) and single nucleotide polymorphisms* (SNP)).

### 2.8. The Importance of Sampling Strategies

The PCE model has been often criticized due to incorrect sampling strategies, for example, that do not take into account the Wahlund effect [27]. This is wrong. In all the data analyzed by us, we have made sure that lack of recombination was not due to a physical isolation by time and/or space.

A two-fold sampling strategy is recommendable to test the PCE model: (1) in close sympatry, including within the same host or vector, in a short time span, to verify that lack of recombination is not explainable by a “starving sex pattern” [6], that is to say: a situation where two different genotypes do not have any opportunity for mating because they are not at the same place at the same time. (2) Over the whole ecogeographical range of the species under study on long periods of time, the stability of the clonal genotypes and of the NCs needs to be tested in space and time.

## 3. *Leishmania* and the PCE Model

The available data, either dealing with high-resolution classic markers such as microsatellites, or with genomic markers, mainly deal with the *Leishmania donovani*/*infantum* species complex.

### 3.1. Widespread Clonal Genotypes

A paradigmatic case is the so-called *L. infantum* MON-1 genotype. “MON” (for the city of Montpellier, Southern France) designates a given MLG characterized by the standard MLEE typing system of the Montpellier Faculty of Medicine, relying on 15 isoenzyme loci and starch gels electrophoresis. By definition, a given MON genotype is therefore a monomorphic MLEE MLG. This is the case for MON-1, which has been repeatedly sampled since the 1980s from various hosts, over vast geographic area, from Latin America (“*L. chagasi*”, which is now synonymous to *L. infantum*) to Europe and North Africa [28,29,30]. We will see further that MON-1 is not a clone, but rather a “clonet”, since it proves to be highly heterogeneous when microsatellite typing, which has a higher resolution than MLEE, is applied (Figure 1).

Widespread MLGs are also observed with microsatellite typing. In South America, a survey of *L. infantum* strains pertaining all to the same zymodeme*, characterized by 14 microsatellite loci, has revealed the presence of a ubiquitous MLG sampled 52 times in 14 Brazilian states and in Paraguay [31]. A similar study dealing with *L. infantum*, performed in Portugal with also 14 microsatellite loci has evidenced a widespread MLG sampled 28 times in dogs, foxes and humans [32]. This microsatellite MLG was identified as MON-1.

An ample survey of *L. infantum* strains, both from the new World and the Ancient world, has sampled ubiquitous microsatellite MLGs on the two sides of the Atlantic Ocean [33]. Again, these microsatellite MLGs pertained to MON-1.

An analysis of 55 *L. infantum* strains from Italy with 15 microsatellite loci, has shown two cases of ubiquitous MLGs. One was isolated three times in dogs from three different regions. The other one was sampled twice: one time in a patient with visceral leishmaniasis, one time in a phlebotomine sandfly from another region.

The species *L. donovani* appears monomorphic in the Indian subcontinent when surveyed by microsatellites and classic Molecular Sequence Typing* (MLST). However, the development of a new MLST scheme shows that this species is genetically heterogeneous in this region of the world [34]. With the new typing approach, two MLGs (sequence types* = STs), namely ST4 and ST5, appear widespread over several Bangladeshi districts.

### 3.2. Deep Phylogenies, Bifurcating Trees, Near-Clades

At the level of the whole species, in all the cases for which convenient data are available, such patterns are abundant in the *Leishmania* genus.

In the complex *L. donovani/infantum/archibaldi*, even with the outdated marker MLEE, that lacks resolution and is subject to homoplasy*, various NCs are evidenced (Figure 1 left, [29]).

With genomic typing, which has a higher resolution than MLEE, both *L. donovani* and *L. infantum* exhibit structuration into several NCs, that are partially, but not totally linked to geographical distance (Figure supp2-V2 in [35]).

The subdivision of *L. infantum* into several NCs is confirmed by microsatellite analysis in Brazil [31], in Europe [28], in Northern Italy [36], in Sicilia [37], in Israel [38], in Tunisia [39], in Armenia [40] and in various other countries [30]. In all the cited studies, bifurcating trees cannot be explained by geographical separation alone, even if this last parameter plays a role (see population A in [37]; see their Figure 5). In [37,40], robustness of the branchings is corroborated by Bayesian, non-phylogenetic methods (STRUCTURE) [20]. In Brazil, NCs are confirmed by phylogeny, STRUCTURE and factorial correspondence analysis [31].

In Bangladesh, NCs are confirmed also in *L. donovani* by microsatellite typing [34].

NCs within *L. braziliensis* are corroborated by MLEE and PCR-RFLP ITSrDNA, a congruence that supports the robustness of these bifurcating trees (test g, [4]).

NCs within the *Leishmania braziliensis/peruviana* complex are corroborated by: (i) the splitstree and STRUCTURE software; (ii) amplified fragment length polymorphism* (AFLP), MLEE, MLST and pulse field gel electrophoresis* (PFGE) [41].

The species *L. tropica* in the Old World exhibits also NCs, that have been stable for the last 55 years [42]. This pattern, revealed by microsatellite typing, is corroborated by Bayesian and phylogenetic analyses. NCs are clearly linked to geographical distance, but not only. Bifurcating trees are visible within each geographical place too (Figure 3 in [42]).

MLEE + MLST typing shows NCs within *L. tropica* and *L. killicki* (Figure 1 in [43]).

### 3.3. Linkage Disequilibrium (LD)

LD is a population genetics statistic that usefully completes the phylogenetic analysis used to evidence bifurcating trees and NCs. If LD is strong, genotypes at all loci are linked, which allows indirect typing: the use of one, or a few, marker(s) makes it possible to characterize the whole MLG. For example, typing with the sole gene k26 makes it possible to characterize the various microsatellite MLGs that pertain to the MLEE MLG MON-1 [37].

A statistically significant LD is evidenced within *L. braziliensis* by microsatellite typing [9].

LD was detected in *L. infantum*, also by microsatellite typing [28]. The authors concluded that the four populations under survey exhibit “a predominantly clonal structure”.

In the same species, in Brazil, out of 2665 SNPs, 75% exhibit LD, whereas 25% do not [44].

In Turkey, within a newly identified MLEE MLG, highly significant LD was evidenced among SNP genotypes. The authors concluded that this population undergoes “mainly clonal reproduction” [45].

In *L. donovani*, in the Indian subcontinent, within a very small subdivision of the species (group of genotypes that are monomorphic with microsatellite typing, although this marker has a fine resolution), “a lack of linkage disequilibrium decay between SNP pairs with genomic distance in the core population reflects a lack of detectable recombination within the six main genetic groups (ISC2-7) across the entire genome” [46].

The congruence between several markers that tag different parts of the genome is a particularly telling case of LD (test g, [4]).

In several species, the MLEE classification of the “MON” system” is corroborated by markers with a higher resolution (microsatellites, MLST, genomic markers), which shows a strong LD between MLEE on the one hand, and these other markers on the other hand. This is the case for *L. donovani*/*infantum* [26,30,35,47], and *L. infantum* [28,32,33,36,37,38,39,40,48].

In *L. donovani/infantum*, congruence is observed between MLEE, MLST, Internal Transcribed Spacer (ITS), mini-exon gene typing and intergenic RFLP of the gp63 gene [26].

In *L. braziliensis*/*peruviana*, there is such a strong congruence between AFLP, MLEE, MLST and PFGE [41].

In *L. donovani*, data from microsatellites, SNP typing and copy number variation* (CNV) are congruent, although with different levels of variation [49,50].

### 3.4. Within-Species Diversity—Russian Doll Patterns (RDPs)

A recurrent criticism of the PCE model states that inhibition of recombination is observed between the main evolutionary lineages/NCs that subdivide the species under study, but that genetic exchange is much more abundant within each of these main NCs [14]. On the contrary, the PCE model in its most recent developments posits that within each of the main NCs, predominant clonality is also observed, with widespread MLGs, LD and lesser NCs, down to a microevolutionary level. This is the “Russian doll model” [24].

In *Leishmania*, Russian doll patterns (RDPs) are observed, not only within the main NCs of the species under study, but also within much tinier subdivisions of the species. Such patterns are widespread in the *L. donovani*/*infantum* complex.

Figure 1 shows an illustrative case of it. The left part shows the main genetic clusters (NCs) within the species *L. archibaldi*, *L. donovani* and *L. infantum*, as evidenced by MLEE analysis. 

The MLEE MLG MON-1 is the most widespread MLG of *L. infantum*. However, from a phylogenetic point of view, it is a tiny subdivision of the species, much smaller than the *L. infantum* NCs. Many studies relying on various markers have confirmed that MON-1 is monophyletic*. The right part of the figure reveals that, by use of microsatellite typing, MON-1 actually shows a considerable genetic diversity and is subdivided into many lesser NCs. This is confirmed by genomic analysis (figure supp2 in [35]). This partitioning of MON-1 into several lesser NCs is confirmed by other microsatellite studies (Figure 2 in [30]; Figure 5 in [37]; Figure 4 in [36] Figure 4 in [31]). In the last study, Brazilian MON-1 strains are subdivided into 3 NCs, corroborated by phylogenetic analysis, Bayesian STRUCTURE software and principal component analysis. One of these NCs is widespread, as well as several microsatellite MLGs. In a recent microsatellite study [40], NCs within MON-1 strains are confirmed by Bayesian STRUCTURE and phylogenetic analyses.

A genomic analysis of Turkish *L. infantum* strains [45] has revealed the existence of a monophyletic lineage, that has been identified as a new MLEE MLG, namely MON-309. SNP within this MLEE MLG polymorphism was high, and a strong LD was observed. The authors concluded that the population structure within this tiny subdivision of *L. infantum* was “primarily clonal”. The proportion of meiosis events was evaluated as 1.3 × 10^−5^/mitose.

*L. donovani* strains from the Indian subcontinent, most of them pertaining to the MLEE MLG MON-2, analyzed with microsatellite typing, show a typical RDP (Figure 1 in [51]), with many lesser NCs corroborated by both phylogenetic analysis (left of the figure) and Bayesian STRUCTURE software (right of the figure). The clustering shows a strong association with geography. However, it is not total, and the RDP still obtains within each of the geographical subdivisions.

A genomic analysis of *L. donovani* from Nepal, Bangladesh and India shows an even more extreme case of RDP [46]: the so-called core 191 genotype is a microsatellite “clonet”, since it is monomorphic with this marker, although microsatellites have a higher power of resolution than MLEE. It can therefore be considered that core 191 is a minuscule phylogenetic subdivision of *L. donovani*. Now, with analysis by WGS and SNPs, the 191 strains of this core group are partitioned into 6 monophyletic lines supported by strong bootstrap values (Figure 2). These lesser NCs are corroborated by both model-based clustering and phylogenetic analysis (maximum likelihood). The ISC5 line shows an additional lesser subdivision. Hybrid lines were recorded, however without disrupting the partitioning into NCs. Within each of the 6 NCs, LD is observed. Time and space scales are worth being noted: most of the 6 NCs exhibited stability from 2002 to 2011, and were isolated both in India and in Nepal. The common ancestor of these 6 NCs can be dated to the mid-XIX° century. Th eISC2 and ISC 4-6 NCs would be as recent as the year 1960. This means that the concept of “measurably evolving pathogens” [52], microbial species which epidemiology can be traced at a microevolutionary level, is applicable to this *Leishmania* species.

### 3.5. Leishmania: Species, or Near-Clades?

*Leishmania* species have been first described on phenotypic characters (“phenospecies”). These phenotypic traits many times concerned pathogenicity. For example, *L. infantum* is the causative agent of visceral leishmaniasis in infants. Subsequently, with the advent of reliable molecular markers, chiefly MLEE, *Leishmania* species were mainly described on phylogenetic criteria (phylogenetic species concept [53]), with limited efforts to look for parity between phylogenetic and phenotypic traits. This led to an inflation of new *Leishmania* “species”. It is remarkable that at the same time, in the late 1970s and early 1980s, pioneering works by Miles et al. [54,55] limited themselves to describe new “zymodemes” (MLEE MLGs) within *T. cruzi*, the agent of the Chagas disease. *Leishmania* specialists could have well described mere zymodemes, and *T. cruzi* specialists could have equally made new species with their zymodemes. From an evolutionary point of view, *T. cruzi* zymodemes and *Leishmania* “species” are perfectly equivalent. Within the PCE framework, both are NCs. Different traditions in different teams are the only explanation for these divergent nomenclatures. A telling comparison has been performed in Michel Tibayrenc’s lab in the 1990s. With exactly the same markers (RAPD) and the same data processing (unweighted Pair-Group Method with Arithmetic Averages dendrograms), the level of genetic divergence in the whole *Leishmania* genus is comparable to the species *Trypanosoma cruzi* and *T. congolense*, and the divergence between the *L*. *braziliensis*/*peruviana* complex and the *L*. *panamensisis*/*guyanensis* complex is similar to the divergence between the lesser NCs within *T. cruzi* (Figure 3). This underlines the arbitrary side of species description, especially in the *Leishmania* genus. The NC concept suggests that in given species, there is a virtually unlimited number of NCs, then lesser NCs, then lesser, lesser NCs, etc. To avoid a distressing inflation of numbers of species, a new species should be described with caution, and should have both a clear phylogenetic individuality (NC) and a relevant epidemiological and/or medical relevance.

It has been proposed to set a lower limit of phylogenetic divergence to allow the description of new *Leishmania* species [57]. This approach is similar to the concept of genospecies in bacteria [58]. This criterion is arbitrary. A double criterion (phylogenetic divergence and genetic discreteness (NC) + epidemiological and/or medical relevance) is preferable.

### 3.6. Widespread Aneuploidy, Population Genetics and the PCE Mode

In the PCE model, as recalled many times, the definition of clonality limits itself to restrained genetic recombination. With this meaning, selfing/inbreeding is only a particular case of clonality. However, some researchers do not accept this definition, and distinguish selfing/inbreeding from “true” (mitotic) clonality [9]. The means to establish the presence of selfing/inbreeding in *Leishmania* are to show that the populations under study exhibit a deficit of heterozygotes*. Mitotic clonality should on the contrary produce an excess of heterozygotes, till a situation of fixed heterozygosity (only heterozygous genotypes) is observed. This approach is based on the hypothesis that *Leishmania* parasites are diploid*. Now, a growing set of evidence suggests that many *Leishmania* species exhibit widespread aneuploidy* [59]. Aneuploidy has been hypothesized ln *L. major*, *L. infantum*, *L. braziliensis* [60], *L. major* [61,62], *L. mexicana*, *L. major*, *L. infantum*, *L. braziliensis* [63].

If *Leishmania* parasites are aneuploid, this makes population genetics tests based on the hypothesis of diploidy [9] questionable [59]. We do not consider that such tests are invalid. However, they should be interpreted with caution, the more so since that strong linkage disequilibrium in parasite natural populations is recorded both with excess and deficit of heterozygotes [64].

It can be noted that aneuploidy is incompatible with classic meiotic mechanisms. On the contrary, it is perfectly compatible with clonal propagation. It is possibly adaptative through a mechanism of gene dosage, favoring drug resistance [63].

Aneuploidy would lead to purging heterozygosity through frequent passages through haploidy* [65], which could explain the frequent heterozygote deficit in *Leishmania*. Other hypotheses to account for heterozygote deficit include homoplasy, large allelic drop out, null alleles and genome-wide mitotic gene conversion [5]. These hypotheses should not be considered separately, as it has been proposed [9] since they are not exclusive of each other [5].

### 3.7. Experimental Mating, Hybrid Lines, Meiosis Genes, and the PCE Model

Successful experimental recombination has been performed in *Leishmania* [27,66]. The frequency of successful hybrid genotypes seems to be low: “2.5 × 10^−5^ or less, after correcting for recovery of only doubly drug–resistant offspring” [66]. In the case of *L. major*, this frequency is lower than 10^−4^ [61]. These experiments suggest meiosis-like mechanisms [27]. The potentiality for mating is indisputable in *Leishmania*. However, experiments convey no information about the frequency of recombination in natural populations and its actual impact on population structure: “Overall, the current debate regarding *Leishmania* reproductive strategies reflects mainly the mode of genetic exchange, its frequency and impact on population structure, not whether or not it occurs” [67]. Successful experimental recombination is, therefore, not a challenge for the PCE model.

This is the same for the presence of hybrid lines in nature. Such hybrids between different species have been recorded between *L. infantum* and *L. major* [68], or within species (*L. infantum* [69]; complex *L. braziliensis*/*panamensis* [70]). In the last study, the hybrids appear to propagate clonally and to be sterile. This situation is quite comparable to the one observed in *Trypanosoma cruzi*, where hybridization is observed between NCs in the work of [71] and within NCs in the work of [72]. As it is the case for the agent of the Chagas disease, hybridization does not disrupt the predominant clonal population structure in *Leishmania*: NCs remain stable in space and time, and discrete: “Different rates of sex and clonality may occur in *L. infantum* depending on demographic or ecological variation within landscapes or between the parasite’s evolutionarily native (Old World) and introduced (New World) range.” [69]. Still, the fact remains that the species *L. infantum* is clearly structured into stable NCs, down to a microevolutionary level (Figure 1).

Lastly, the presence of meiosis genes has been recorded in many parasitic species, including *Giardia*, *Trichomonas* and *Entamoeba* [73]. Meiosis genes are present also in *L. major* and *L. donovani* [74]. If these genes really permit meiosis, it says nothing about the frequency of this in natural populations. Moreover, “meiosis genes” may have different functions through an exaptation* process: “Evolution is constantly re-using old genes for new purposes” [75].

## 4. Conclusions

In all the cases for which convenient data are available, the *Leishmania* genus perfectly fits all the expectations of the PCE model. Russian doll patterns are especially telling in this genus, since they are verified, not only within the main NCs that subdivide *Leishmania* species, but also within individual MLEE MLGs (Figure 1), and even individual microsatellite MLGs (Figure 2). So, in *Leishmania* parasites, the PCE patterns are verified down to a microevolutionary level, at the scale of “measurably evolving pathogens” [52]. 

Meiosis and hybridization events certainly play an important role at an evolutionary level in the *Leishmania* genus, as it is the case also for *T. cruzi*, the agent of Chagas disease. However, their impact is not sufficient to break up the prevalent pattern of clonal population structure. Bifurcating trees and NCs remain persistent.

The population structure of *Leishmania* is not unique. Very similar evolutionary patterns are observed in other parasites (*T. cruzi*, *T. congolense*), in yeasts (*Cryptococcus neoformans*), and also, various bacteria species [5,6,7]. The double strategy of clonal propagation and of occasional mating may be a typical evolutionary trait of adaptation to the parasitic life.


***Glossary of specialized terms:**


Amplified fragment length polymorphism (AFLP): Selective amplification of genomic restriction fragments (obtained by RFLP) by PCR using randomly selected primers.

Aneuploidy: When, in a given species, different chromosomes may have different copy numbers.

Bootstrap analysis: In phylogenetic analysis, generation of pseudoreplicate datasets by random sampling of the original character matrix to create new matrices that have the same size than the original. The frequency with which a given branch is reproduced by this randomization procedure is recorded as the bootstrap proportion. These proportions can be used as a measure of the robustness of individual branches of the tree. A bootstrap value of 70 for a given branch means that this branch has been found 70 times out of 100 by the procedure.

Classic marker: Those markers that were widely used before the era of wide use of genomic data. They include multilocus enzyme electrophoresis (MLEE), multilocus sequence typing (MLST), among others.

Copy-number variant, copy-number variation (CNV): DNA sequence ≥1 kb, present with a variable copy number by comparison with a reference genome. Includes insertions, deletions and duplications.

Diploidy: Two copies of each chromosome in a given organism.

Exaptation: The process by which a given trait acquires functions for which it was not originally adapted. For example, the bat’s arms have become wings.

Genetic recombination: Exchange of genetic material occurring at two or more different genetic loci between different individuals, which leads to the production of offspring with combinations of genetic characters that differ from those observed in either parent.

Haploidy: Only one copy of each chromosome in a given organism.

Heterozygote, heterozygous: If the two alleles in a given individual of a diploid species are genetically identical, the individual is termed “homozygote.” If the two alleles are different, the individual is “heterozygote.”

Homoplasy: Possession in common by distinct phylogenetic lineages of identical characters that do not come from a common ancestry. The origin of homoplasic characters include: (*a*) convergence (possession of identical characters derived from different ancestral characters, due to convergent evolutionary pressure; for example: wings of bats on the one hand, and of birds on the other hand), (*b*) parallelism (possession of identical characters derived from a same ancestral character, and independently generated in different phylogenetic lines), and (*c*) reversion (restoration of an ancestral character from a derived character). In the case of molecular markers, since the space available on a given electrophoresis gel is limited, characters may migrate at the same level while they are coded by different DNA sequences.

Microevolution: Classically: evolution at the level of a given population. More specifically, in the present article: a scale of time that could go down to recent years or at least historical times, a definition that is more convenient to the epidemiology of microbial pathogens and to the testing of the Russian doll model (see text).

Microsatellite: A short DNA sequence, usually 1–4 bp long, that is repeated together in a row along the DNA molecule. In humans, as in many other species, there is great variation in the number of repeats from one individual to another and among different populations. Numbers of repeats for a given locus define microsatellite alleles.

Monophyletic: An evolutionary lineage that has a unique common ancestor.

Multilocus Enzyme Electrophoresis (MLEE): Isoenzymes are protein extracts from given samples, for example, various pathogen isolates. They are separated by electrophoresis. The gel is then subjected to a histochemical reaction involving the specific substrate of a given enzyme, and the zone of activity of this enzyme is specifically stained. The same enzyme from different samples may migrate at different rates that are the manifestation of genetic differences in the genes coding for these enzymes. These different electrophoretic forms of the same enzyme are referred to as isoenzymes or isozymes.

Multilocus genotype (MLG): The combined genotype of a given isolate or a given individual established with several genetic loci.

Multilocus Sequence Typing (MLST): Method of pathogen characterization based on the sequencing of several housekeeping genes.

Panmixia, panmictic: The situation where genetic exchange occurs at random in a given population. Panmictic expectations are the verification of this state by various population genetics tests.

Pulsed field gel electrophoresis: Technique used for separating large DNA fragments by applying to a gel matrix an electric field that periodically changes direction.

Random primed amplified polymorphic DNA (RAPD): Method of DNA characterization, also known as Arbitrarily-Primed Polymerase Chain Reaction or AP-PCR. In the classical Polymerase Chain Reaction (PCR) method, the primers used are known DNA sequences, whereas the RAPD technique relies on primers whose sequence is arbitrarily determined.

(Genetic) Recombination: Exchange of genetic material occurring at two or more different genetic loci between different individuals, which leads to the production of offspring with combinations of genetic characters that differ from those observed in either parent.

Restriction fragment length polymorphism (RFLP): Variability in the DNA of a given organism revealed by the use of restriction endonucleases. The endonuclease cuts the DNA at specific restriction sites, and the polymorphism of the DNA fragments so obtained can be visualized on gels.

Sequence type (ST): Individual MLG identified by MLST.

Single nucleotide polymorphisms (SNP): Polymorphisms of one-letter variations in the DNA sequence.

Wahlund effect: Classically, in population genetics, the Wahlund effect designates a situation where a heterozygote deficit is due to the fact that two distinct populations with different allelic frequencies, between which physical obstacles (time and/or space) obtain, have been mistakenly plotted into a unique population. Here, the Wahlund effect is understood in a broader meaning, when any apparent departures from panmictic expectations are explained only to physical obstacles (space and/or time) to gene flow.

Whole genome sequencing (WGS): The sequencing of the complete genome, including noncoding sequences.

Zymodeme: A set of pathogen strains that share the same MLEE MLG.

## Figures and Tables

**Figure 1 microorganisms-09-02409-f001:**
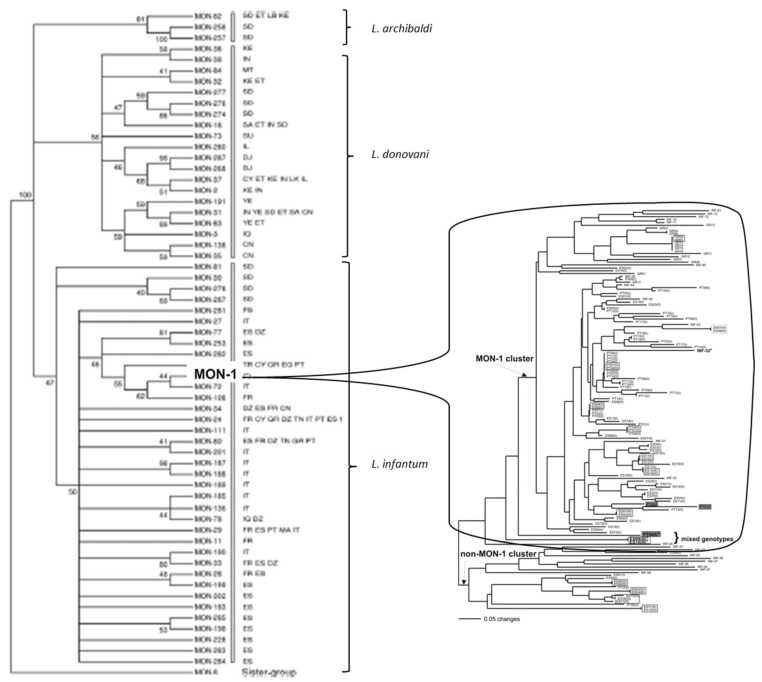
Left: various NCs evidenced within the *L. donovani*/*infantum* complex by MLEE analysis (after [29], copyright permission was obtained from Cambridge University Press). The MON-1 MLEE MLG is only one tiny part of one of the lesser NCs within the species *L. infantum*. Right: microsatellite analysis, that has a higher resolution power than MLEE, shows that MON-1 is genetically heterogeneous and exhibits various lesser near-clades: this is a typical Russian doll pattern (after [28]).

**Figure 2 microorganisms-09-02409-f002:**
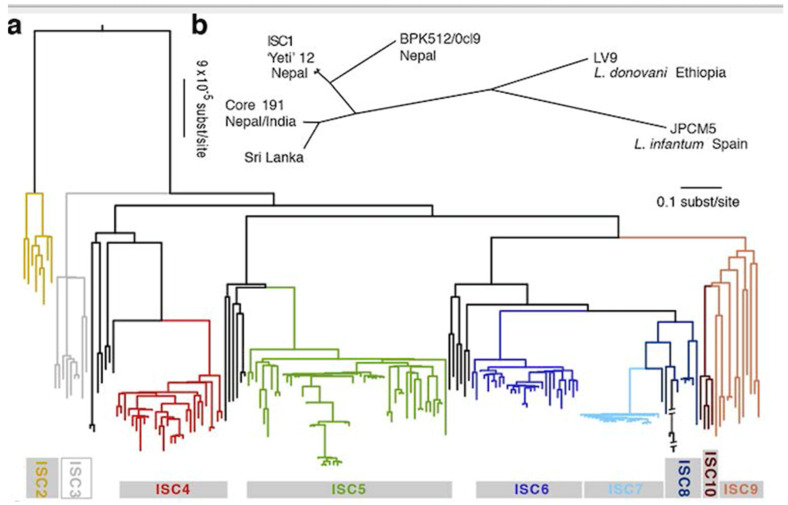
Various lesser NCs (ISC1 to 10) evidenced within one microsatellite MLG (Core 191 in Nepal and India, top right of the figure) of the L. donovani/infantum complex by genomic analysis. a = SNP-based phylogenetic tree. b = Unrooted phylogenetic network of the complex based on genetic distances between isolates described here (after [46]).

**Figure 3 microorganisms-09-02409-f003:**
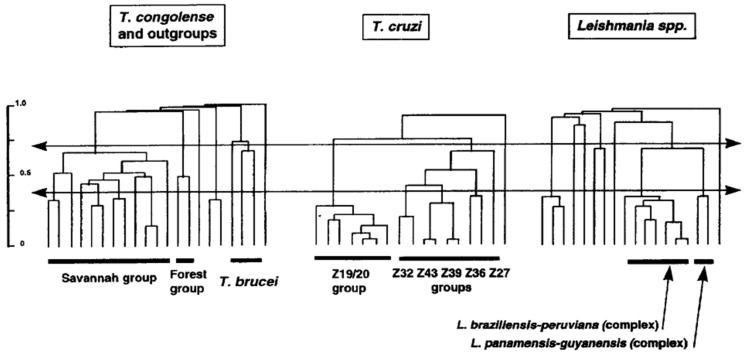
Comparisons of the evolutionary distances in the whole genus *Leishmania* and in the species *Trypanosoma congolense* and *Trypanosoma cruzi* (after [56], copyright permission was obtained from Elsevier).

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
