# Peer review of "Leishmania and the Model of Predominant Clonal Evolution"

_microorganisms, 2021, doi:10.3390/microorganisms9112409_

Round 1
Reviewer 1 Report
The manuscript “Leishmania and the model of predominant clonal evolution” shows a interesting debate about the recent development of the theory “the clonal theory of parasitic protozoa”, dealing with the Leishmania genus. The topic has been extensively covered in a critical and satisfactory manner. It may be published in Microorganism. Need to adjust figure 1 and figure 2 which are not visible clearly.
Line 105 – change side in Side
Line 155 – space in figure 1
Line 206 – h26 or k26?
Author Response
Thank you for your careful review of our manuscript. Unfortunately, we cannot make any better for figures 1 and 2.
Reviewer 2 Report
The work of Tibayrenc and Ayala is a review and it must be clearly labeled as such. I liked the way it is presented and recommend acceptance of this manuscript. I also appreciate that authors discussed recent -omics data. A few minor suggestions:
- Please fix the references. Their style is not unified and this makes a bad impression.
- Please discuss the recent exploration into L. tropica population structure reported here: doi:10.1016/j.actatropica.2021.105888
Author Response
Thank you for your suggestions. We have carefully reviewed the list of references, which, as a matter of fact, was not homogeneous. We have corrected all this.
As for the paper by Charyyeva et al., it is very interseting, however it is not suitable for our approach, that must rely on the analysis of genes that segregate independently. If this condition is not fulfilled, it is impossible to explore segregation and recombination. Internal transcribed spacer 1 (ITS1) sequences are a family of genes that are linked and do not segregate independently. See a similar case in: Tibayrenc, M. & Ayala, F.J. 2019. Are the multiple Trypanosoma cruzi infections in Louisiana rodents caused by independent genetic clones? J. Microbiol., Immunol. & Infection 53: 668-669 https://doi.org/10.1016/j.jmii.2019.04.014